# Molecular Pathogenesis of the Coronin Family: *CORO2A* Facilitates Migration and Invasion Abilities in Oral Squamous Cell Carcinoma

**DOI:** 10.3390/ijms222312684

**Published:** 2021-11-24

**Authors:** Ikuko Kase-Kato, Shunichi Asai, Chikashi Minemura, Kenta Tsuneizumi, Sachi Oshima, Ayaka Koma, Atsushi Kasamatsu, Toyoyuki Hanazawa, Katsuhiro Uzawa, Naohiko Seki

**Affiliations:** 1Department of Oral Science, Graduate School of Medicine, Chiba University, Chiba 260-8670, Japan; kato.ikuko@chiba-u.jp (I.K.-K.); minemura@chiba-u.jp (C.M.); tsuneizumikenta@chiba-u.jp (K.T.); Sachi.o8952@chiba-u.jp (S.O.); axna4812@chiba-u.jp (A.K.); kasamatsua@faculty.chiba-u.jp (A.K.); uzawak@faculty.chiba-u.jp (K.U.); 2Department of Functional Genomics, Graduate School of Medicine, Chiba University, Chiba 260-8670, Japan; cada5015@chiba-u.jp; 3Department of Otorhinolaryngology/Head and Neck Surgery, Graduate School of Medicine, Chiba University, Chiba 260-8670, Japan; thanazawa@faculty.chiba-u.jp

**Keywords:** oral squamous cell carcinoma, coronin, *CORO2A*, microRNA, *miR-125b-5p*, *miR-140-5p*

## Abstract

In humans, the coronin family is composed of seven proteins containing WD-repeat domains that regulate actin-based cellular processes. Some members of the coronin family are closely associated with cancer cell migration and invasion. The Cancer Genome Atlas (TCGA) analysis revealed that *CORO1C*, *CORO2A*, and *CORO7* were significantly upregulated in oral squamous cell carcinoma (OSCC) tissues (*p* < 0.05). Moreover, the high expression of *CORO2A* was significantly predictive of the 5-year survival rate of patients with OSCC (*p* = 0.0203). Overexpression of *CORO2A* was detected in OSCC clinical specimens by immunostaining. siRNA-mediated knockdown of *CORO2A* suppressed cancer cell migration and invasion abilities. Furthermore, we investigated the involvement of microRNAs (miRNAs) in the molecular mechanism underlying *CORO2A* overexpression in OSCC cells. TCGA analysis confirmed that tumor-suppressive *miR-125b-5p* and *miR-140-5p* were significantly downregulated in OSCC tissues. Notably, these miRNAs bound directly to the 3′-UTR of *CORO2A* and controlled *CORO2A* expression in OSCC cells. In summary, we found that aberrant expression of *CORO2A* facilitates the malignant transformation of OSCC cells, and that downregulation of tumor-suppressive miRNAs is involved in *CORO2A* overexpression. Elucidation of the interaction between genes and miRNAs will help reveal the molecular pathogenesis of OSCC.

## 1. Introduction

Oral squamous cell carcinoma (OSCC), which originates from oral keratinocytes in the oral cavity, accounts for 40% of head and neck squamous cell carcinomas (HNSCCs) [1]. According to Global Cancer Statistics 2018, there were approximately 350,000 new cases with OSCC and 180,000 deaths from the disease per year worldwide [2]. Most OSCC cases are detected at an advanced stage, and the 5-year survival rate of patients with advanced OSCC is approximately 50% [3]. Advanced-stage patients are treated with combination therapies, such as surgery, radiation, chemotherapy, and immunotherapy [4]. However, OSCC cells acquire resistance to these treatments, resulting in local recurrence and distant metastasis [5]. Unfortunately, there is no effective treatment for patients who have acquired treatment resistance [6]. Elucidation of the molecular mechanisms of acquiring treatment resistance and subsequent development of distant metastasis is indispensable for the development of new treatments for OSCC.

Metastasis is the process by which cancer cells migrate to distant sites via the lymphatic system or bloodstream to form colonies, and it is an important event determining the prognosis of patients [7]. Various molecules play roles in the metastatic process in a complex manner. Cancer cells induce changes in cell adhesion and degradation of surrounding cells and acquire the ability to migrate within tissues and metastasize [8]. The dynamic involvement of actin is required for cancer cells to alter their morphology and promote migration and invasion [9]. At the edge of migrating and invading cells, actin polymerization is controlled by various actin regulators, e.g., Rho GTPases, WASP and WAVE proteins, and actin-related proteins [10].

The coronin family comprises WD-repeat proteins, which are expressed in a large number of eukaryotic organisms [11]. The WD domain is thought to function as a stable platform for interacting with other proteins [12]. As the typical structure of a coronin protein, it contains three to five WD-repeat clusters forming the central core domain, as well as a coiled-coil domain in the carboxy terminus [13]. In humans, seven coronin genes have been identified: *CORO1A* (coronin 1), *CORO1B* (coronin 2), *CORO1C* (coronin 3), *CORO2A* (coronin 4), *CORO2B* (coronin 5), *CLIPINE* (coronin 6), and *POD1* (coronin 7) [14].

Previous studies showed that coronins are associated with the actin-related protein 2/3 complex and are involved in an F-actin rearrangement, suggesting that coronins act as actin-binding proteins [15]. Overexpression of *CORO1C* in cancer cells has been reported in a wide range of cancers, and its overexpression contributes to cancer cell migration and invasion [16]. In this study, we investigated the clinical significance of human coronins using The Cancer Genome Atlas (TCGA) database. Analysis of the TCGA-HNSC database showed that expression of *CORO2A* was upregulated in HNSCC tissues, and its high expression significantly predicted the 5-year survival rate of patients with HNSCC.

A large number of studies have shown that microRNAs (miRNAs) act as pivotal controllers of gene expression [17]. miRNAs, short single-stranded molecules, negatively control gene expression in normal and diseased cells in a sequence-dependent manner [18,19]. In cancer cells, a vast number of studies have demonstrated that downregulation of tumor-suppressive miRNAs caused overexpression of oncogenes, and these events facilitated cancer cell aggressiveness, e.g., proliferation, metastasis, and drug resistance [20,21].

The study aimed to investigate the oncogenic function of *CORO2A* and to clarify the involvement of miRNAs that control *CORO2A* expression in OSCC cells.

## 2. Results

### 2.1. Expression and Clinical Significance of the Coronin Family in Patients with OSCC According to TCGA Database Analysis

Expression levels of the coronin family members (*CORO1A*, *CORO1B*, *CORO1C*, *CORO2A*, *CORO2B*, *CORO6*, and *CORO7*) were evaluated using the TCGA database (TCGA-OSCC). The expression levels of three coronin genes (*CORO1C*, *CORO2A*, and *CORO7*) were significantly upregulated in OSCC tissues (*n* = 313) compared with normal tissues (*n* = 30) (Figure 1). On the other hand, the expression of *CORO2B* was significantly downregulated in OSCC tissues (Figure 1). There was no significant difference in the expression levels of the other three genes (*CORO1A*, *CORO1B*, and *CORO6*) between OSCC and normal tissues (Figure 1).

To determine the clinical effectiveness, a clinicopathological analysis of the coronin family was performed using TCGA-OSCC data. Patients with high expression of *CORO1B* and *CORO2A* had a significantly worse prognosis compared with those with low expression (Figure 2A and Appendix A). Multivariate Cox regression analysis was performed about expression levels of *CORO1B* and *CORO2A* with other expected prognostic factors (age, disease stage, and pathological grade). As result, *CORO2A* expression levels were independent prognostic factors (Figure 2B).

Based on the results of these analyses, CORO2A was selected among the coronin family members for the functional analyses.

### 2.2. Overexpression of CORO2A in OSCC Clinical Specimens

Expression of the *CORO2A* protein was investigated by immunostaining in OSCC clinical specimens. Aberrant expression of *CORO2A* was detected in OSCC lesions (Figure 3). In contrast, there was almost no *CORO2A* expression in the normal epithelium (Figure 3). Clinical features of 4 OSCC cases used for immunohistochemical staining were summarized in Appendix A.

### 2.3. Effects of CORO2A Knockdown on the Proliferation, Invasion, and Migration of OSCC Cells

To assess the oncogenic function of *CORO2A* in OSCC cells, we performed knockdown assays using small interfering RNAs (siRNAs). Prior to this experiment, we evaluated the expression of *CORO2A* in OSCC cells lines (HSC-2, HSC-3, SAS, Sa3, Ca9-22, and Ho-N-1), and found that *CORO2A* expression was detected in all of these cell lines (Figure 4). On the other hand, its expression in normal fibroblast lines (IMR-90 and MRC-5) was weaker compared with the OSCC cell lines (Figure 4).

Next, the inhibitory effect of two different siRNAs targeting *CORO2A* (*siCORO2A-1* and *siCORO2A-2*) on *CORO2A* expression was examined. The *CORO2A* mRNA and protein levels were effectively suppressed by transfection of both siRNAs into SAS and HSC-3 cells (Figure 5A,B).

Knockdown of *CORO2A* had a slight inhibitory effect on cell proliferation in SAS and HSC-3 cells (Figure 6A), and the cell invasion and migration abilities were significantly inhibited after *siCORO2A* transfection in SAS and HSC-3 cells (Figure 4B,C, Appendix A and Appendix A).

### 2.4. Selection of miRNAs That Regulate CORO2A Expression in OSCC Cells

Downregulation of some miRNAs is associated with overexpression of *CORO2A* in OSCC cells. We searched for miRNAs that negatively regulate *CORO2A* expression in OSCC cells. The strategy for identifying miRNAs that regulate *CORO2A* expression is shown in Figure 7A. An analysis of the TargetScan database (release 7.2) combined with our miRNA expression signature of OSCC (accession number: GSE184991) revealed that three miRNAs (*miR-125a-5p*, *miR-125b-5p*, and *miR-140-5p*) regulate *CORO2A* expression in OSCC cells (Figure 7B). TCGA database analysis showed that the expression levels of *miR-125b-5p* and *miR-140-5p* were significantly reduced in OSCC tissues (*n* = 297) compared with normal tissues (*n* = 30) (Figure 7C). Based on these results, we investigated the regulation of *CORO2A* expression by *miR-125b-5p* and *miR-140-5p*.

### 2.5. Regulation of CORO2A Expression by miR-125b-5p and miR-140-5p in OSCC Cells

Both the mRNA and protein levels of *CORO2A* were reduced by *miR-125b-5p* transfection in SAS and HSC-3 cells (Figure 8A,B). To investigate whether *miR-125b-5p* binds directly to *CORO2A* in OSCC cells, we conducted a dual-luciferase reporter assay. Luciferase activity was significantly reduced following co-transfection with miR-125b-5p and a vector containing the *miR-125b-5p*-binding site of *CORO2A*. On the other hand, co-transfection with a vector lacking the *miR-125b-5p*-binding site of *CORO2A* resulted in no change in luciferase activity (Figure 8C).

Both the mRNA and protein levels of *CORO2A* were reduced by *miR-140-5p* transfection in SAS and HSC-3 cells (Figure 9A,B). A dual-luciferase reporter assay showed that luciferase activity was significantly reduced following co-transfection with *miR-140-5p* and a vector containing the *miR-140-5p*-binding site of *CORO2A*. There was no change in luciferase activity following co-transfection with *miR-140-5p* and a vector lacking the *miR-140-5p*-binding site (Figure 9C).

These findings suggest that *miR-125b-5p* and *miR-140-5p* directly regulate *CORO2A* expression in OSCC cells.

### 2.6. Effects of Ectopic Expression of miR-125b-5p and miR-140-5p in OSCC Cells

The tumor-suppressive activities of *miR-125b-5p* and *miR-140-5p* were assessed by ectopic expression of mature miRNAs in SAS and HSC-3 cells. The results showed that cell proliferation was suppressed by *miR-125b-5p* transfection in OSCC cells (Figure 10A). Especially, cancer cell invasion and migration abilities were markedly suppressed by the expression of *miR-125b-5p* in OSCC cells (Figure 10B,C, Appendix A and Appendix A.)

Similar to *miR-125b-5p*, ectopic expression of *miR-140-5p* attenuated OSCC cell aggressiveness, i.e., cell proliferation, invasion, and migration abilities (Figure 11A–C). These findings suggest that *miR-125b-5p* and *miR-140-5p* act as tumor-suppressive miRNAs in OSCC cells. (Appendix A).

## 3. Discussion

The prognosis of patients with OSCC depends largely on the presence or absence of metastasis. Metastasis (regional lymph node or distant metastasis) is a major cause of death in patients with OSCC, and the 5-year survival rate of OSCC patients with metastasis is less than 40% [22]. Searching for molecular networks involved in metastasis is an essential challenge in improving the prognosis of patients with OSCC.

Cancer cells acquire cell motility via regulation of the cytoskeletal structure, allowing them to move travel from the primary tumor site to distant tissues [23]. Cancer cells control F-actin filaments at the leading edge to form various protrusion processes, e.g., lamellipodia, filopodia, and invadopodia [24]. Various actin-related proteins (e.g., Arp2/3, WASP/WAVE, fascin, and tropomyosins) are involved in the formation of these protrusions in a complex manner [25,26,27]. For instance, cortactin (*CTTN*) has multiple binding domains that can bind several proteins, e.g., Arp2/3 complex, F-actin, WASL, WIPE-1, and Src [28]. Overexpression of *CTTN* was reported by several cancers, including OSCC, and its expression enhanced cancer cell aggressiveness in OSCC [29,30]. Moreover, aberrant expression of *CNNT* significantly predicted the HNSCC prognosis [31].

Coronins are actin-binding proteins containing WD repeats that are evolutionarily conserved from invertebrates to vertebrates [13]. In humans, the coronin family comprises at least seven genes [14]. Among these, *CORO1C* contributes to invadopodium formation via the F-actin and Arp2/3 complex [32]. Overexpression of *CORO1C* has been reported in various solid tumors, including glioblastoma, hepatocellular, breast, lung, gastric, and colorectal cancers [32,33,34,35,36,37]. In colorectal cancer, high expression of *CORO1C* is associated with malignant phenotypes, such as lymph node and distant metastases. Moreover, the oncogenic PI3K/AKT signaling pathway was suppressed by the knockdown of *CORO1C* [38]. In gastric cancer, the knockdown of *CORO1C* markedly suppressed the malignant transformation of cancer cells [37].

*CORO2A* is a component of the nuclear receptor co-repressor complex involved in the actin-dependent activation of inflammatory response genes [39]. There have not been many reports on *CORO2A* expression in human cancers. A previous study showed elevated expression of *CORO2A* in colorectal carcinoma tissues, and a link between its expression and oncogenic MAPK14 and PRMT5 signaling pathways [40]. A recent study showed that expression of *CORO2A* was associated with overall survival and relapse-free survival in patients with triple-negative breast cancer; moreover, knockdown of *CORO2A* reduced malignant transformation and induced cell cycle arrest [41]. The results of that study are consistent with our results in OSCC showing that *CORO2A* expression is a potential predictor of OSCC prognosis. Furthermore, controlling *CORO2A* expression and *CORO2A*-mediated oncogenic pathways may provide promising therapeutic strategies for OSCC treatment. A large number of cohort studies by other institutions are essential for *CORO2A* expression to gain clinical significance in OSCC.

In this study, we investigated the molecular mechanisms underlying the aberrant expression of *CORO2A* in oral cancer, focusing on miRNAs. A vast number of studies demonstrated that miRNAs act as fine-tuners of gene expression in a sequence-dependent manner [18,19]. Tumor-suppressive miRNAs are frequently downregulated, whereas their oncogene targets are upregulated, in cancer cells. A previous analysis of our RNA-sequencing-based miRNA signature of OSCC revealed that *miR-125b-5p* and *miR-140-5p* were downregulated in cancer tissues [42]. Our present study showed that these miRNAs have tumor-suppressive roles, and *miR-125b-5p* and *miR-140-5p* directly regulate the expression of *CORO2A* in OSCC cells.

Downregulation and antitumor roles of *miR-125b-5p* have been reported in several types of cancers [43,44,45,46,47,48]. For instance, *miR-125b-5p* is downregulated in esophageal squamous cell carcinoma, and ectopic expression assays demonstrated that its expression attenuated cancer cell aggressiveness via regulation of cell cycle regulatory genes and epithelial–mesenchymal transition (EMT)-related genes [49]. In laryngeal squamous cell carcinoma cells, ectopic expression of *miR-125b-5p* blocked glucose consumption by targeting hexokinase-2 [46].

Previous studies have reported downregulation of *miR-140-5p* in several cancer types [50,51,52,53,54]. In hypopharyngeal squamous cell carcinoma, expression of *miR-140-5p* suppressed cancer cell migration and invasion abilities by regulating ADAM10-mediated Notch1 signaling. Moreover, downregulation of *miR-140-5p* was associated with tumor classification and lymph node metastasis [55]. In OSCC cells, overexpression of the long noncoding RNA HCP5 promoted cancer cell proliferation and EMT by adsorbing *miR-140-5p*. Notably, suppression of *miR-140-5p* expression alleviated repression of *SOX4*, a master regulator of EMT in OSCC cells [56].

The finding that the antitumor miRNAs *miR-125b-5p* and *miR-140-5p* are involved in the regulation of *CORO2A* expression in OSCC cells is interesting and new. Detailed functional analyses of *CORO2A* and its regulated antitumor miRNAs will provide important information for elucidating the molecular pathogenesis of OSCC.

In this study, we focused on *CORO2A* among the coronin family and analyzed its oncogenic functions and epigenetic modification in OSCC cells. For other coronin families that have not been analyzed in this study (especially, *CORO1C*, *CORO2B*, and *CORO7* whose expressions are dysregulated in cancer tissues), it is essential to elucidate the molecular pathogenesis of OSCC. Continued analysis of these genes may reveal therapeutic targets for OSCC.

## 4. Materials and Methods

### 4.1. Human OSCC Cell Lines

The six OSCC-derived cell lines (HSC-2, HSC-3, SAS, Sa3, Ca9-22, and HO-1-N-1) and two human fibroblast lines (IMR-90 and MRC-5) were used in this study (Appendix A). These cell lines were obtained from the RIKEN BioResource Center (Tukuba, Ibaraki, Japan).

### 4.2. RNA Extraction and qRT-PCR

RNA was extracted from cell lines and subjected to this study as described previously [42,57,58]. The TaqMan probes and primers used in this study are listed in Appendix A.

### 4.3. Transfection of siRNAs and miRNAs into HNSCC Cells

Transfection of siRNAs and miRNAs into OSCC cell lines was performed using Lipofectamine RNAiMAX reagent (Invitrogen, Carlsbad, CA, USA) according to our previous studies [42,57,58]. The reagents used in this study are listed in Appendix A.

### 4.4. Functional Assays (Cell Proliferation, Migration, and Invasion Assays) in HNSCC Cells

The XTT assay for cell proliferation and the Matrigel chamber assay for the invasion were performed in OSCC cells as described previously [42,57,58]. In the wound healing assay for migration, a wound was created using a micropipette tip in the middle of each plate after siRNA or pre-miRNAs transfection 48 h. We incubated plates at 37 °C at 5% carbon dioxide with a free-serum medium, and live-cell migration was captured after 12 and 24 h.

### 4.5. Clinical Significance of CORO2A in OSCC Patients Based on TCGA-HNSC Data

Gene expression analysis for each gene were obtained from OncoLnc (http://www.oncolnc.org, accessed on 20 April 2021) [59]. For the Kaplan–Meier plot, log-rank test, and Cox proportional hazards regression test, we used TCGA-HNSC clinical data (TCGA, Firehose Legacy) obtained from cBioportal (https://www.cbioportal.org, accessed on 10 April 2020). Among TCGA-HNSC, those whose primary site was in the oral cavity (alveolar ridge, buccal mucosa, floor of mouth, hard palate, lip, and oral tongue) were narrowed down as TCGA-OSCC. The clinical features of TCGA-OSCC (*n* = 343) was shown in Appendix A. In multivariate analysis, gene expression levels, tumor stage, pathological grade, and age at diagnosis were used as covariates.

For these analyses, we used JMP Pro 15.0.0 (SAS Institute Inc., Cary, NC, USA).

### 4.6. Identification of CORO2A Expression Controlled miRNAs

The strategy used to identify *CORO2A* target miRNAs is presented in Figure 7. We selected putative miRNA target sites within the *CORO2A* sequence using TargetScanHuman ver. 7.2 (http://www.targetscan.org/vert_72/, accessed on 10 July 2020) [60]. The expression signature of OSCC miRNA was used for screening. Our OSCC miRNA signature was deposited in the GEO database (accession number: GSE184991). Clinical features of 3 OSCC cases used for miRNA sequence were summarized in Appendix A.

### 4.7. Western Blotting and Immunohistochemistry

The procedure of Western blotting and immunohistochemistry were described as previously [42,57,58]. The antibodies used in this study are shown in Appendix A. Full blots of the membrane are shown in Appendix A

### 4.8. Clinical Specimens

The clinical information of the patients using immunostaining are shown in Appendix A. Our study has been approved by the Ethics Committee of Chiba University (approval number; 28–65, 10 February 2015). The research methodology is implemented in accordance with the standards set by the Declaration of Helsinki.

### 4.9. Plasmid Construction and Dual-Luciferase Reporter Assays

Plasmid vectors containing *CORO2A* with the wild-type sequences of the miRNAs (*miR-125b-5p* and *miR-140-5p*) binding sites in the 3′-UTR and without those sequences were prepared. We have described the methods for transfection and dual-luciferase reporter assays in our previous studies [42,57,58]. The reagents used in this study are listed in Appendix A.

### 4.10. Statistical Analysis

We performed statistical analyses using JMP Pro 15 (SAS Institute Inc., Cary, NC, USA). Differences between the two groups were evaluated using Welch’s t-test. Dunnett’s test was used for multiple group comparisons. A *p*-value less than 0.05 was considered statistically significant.

## 5. Conclusions

This large cohort analysis revealed that high expression of *CORO2A* in OSCC clinical tissues is highly predictive of a worse prognosis in OSCC patients. The knockdown assays suggested that the expression of *CORO2A* facilitates cancer cell malignant transformation, e.g., cell proliferation, migration, and invasion. Tumor-suppressive *miR-125-5p* and *miR-140-5p* directly regulate *CORO2A* expression in OSCC cells. *CORO2A-* and *CORO2A*-mediated oncogenic signaling pathways may provide novel information regarding OSCC molecular pathogenesis.

## Figures and Tables

**Figure 1 ijms-22-12684-f001:**
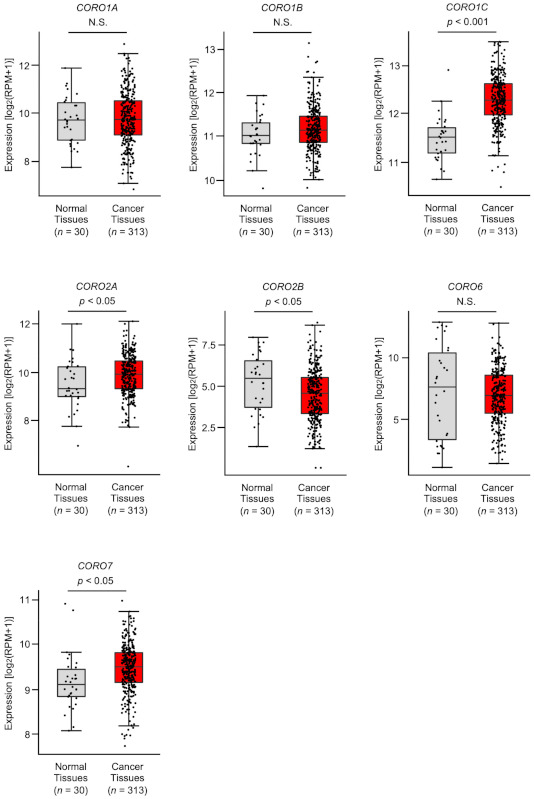
Expression of all members of coronin family by TCGA-OSCC analysis. Expression levels of *CORO1A*, *CORO1B*, *CORO1C*, *CORO2A*, *CORO2B*, *CORO6,* and *CORO7* in OSCC tissues. A total of 313 OSCC tissues and 30 normal epithelium tissues were analyzed (N.S., not significant).

**Figure 2 ijms-22-12684-f002:**
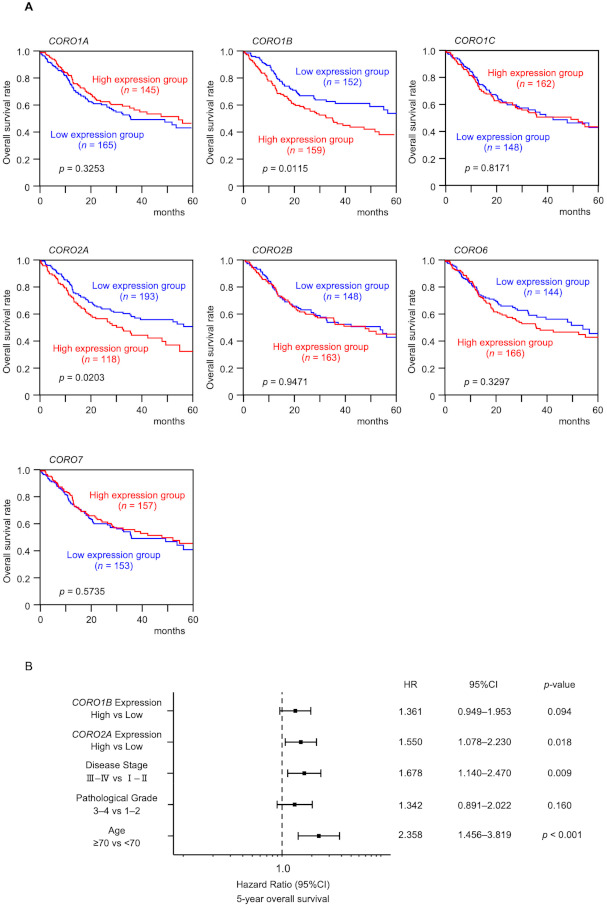
Clinical significance of all members of the coronin family by TCGA-OSCC analysis. (**A**) Kaplan–Meier survival curve analyses of patients with OSCC using data from The Cancer Genome Atlas (TCGA) database. Patients were divided into high and low expression groups according to miRNA expression (based upon median expression). The red line shows the high expression group, and the blue line shows the low expression group. (**B**) Forest plot presenting the results of a multivariate Cox regression analysis of the prognostic value of *CORO1B* and *CORO2A* identified in an OSCC dataset from TCGA (HR: hazard ratio, CI: confidence interval). The expression level of *CORO2A* was determined to be independent prognostic factors in terms of the 5-year overall survival rate after adjustments for tumor stage, age, and pathological grade (*p* < 0.05).

**Figure 3 ijms-22-12684-f003:**
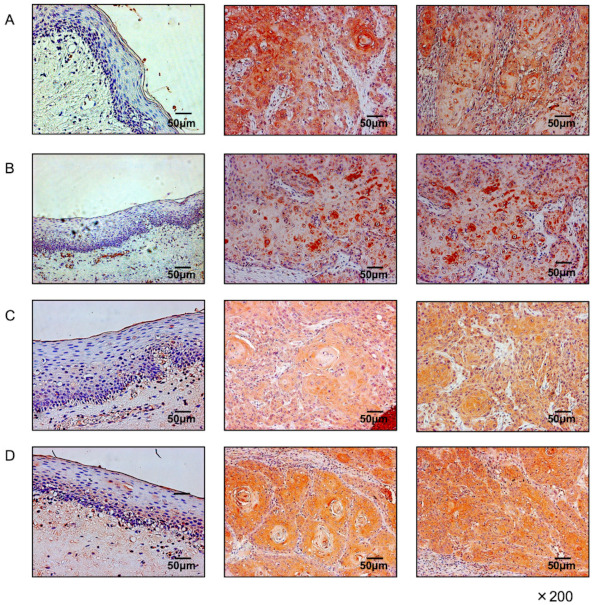
*Overexpression of CORO2A in OSCC clinical specimens*. Immunohistochemical staining of *CORO2A* in OSCC clinical specimens. High expression of *CORO2A* was detected in the nuclei and/or cytoplasm of cancer cells ((**A**–**D**) center and right side) and weak expression in the normal oral mucosa ((**A**–**D**) left side).

**Figure 4 ijms-22-12684-f004:**
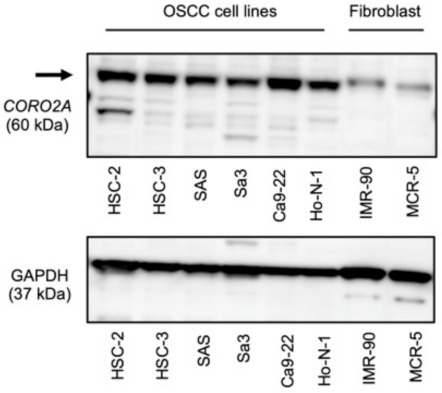
Expression levels of *CORO2A* in the OSCC cell lines. Expression levels of *CORO2A* in OSCC cell lines and normal fibroblast cell lines were evaluated with Western blotting. GAPDH was used as the internal control.

**Figure 5 ijms-22-12684-f005:**
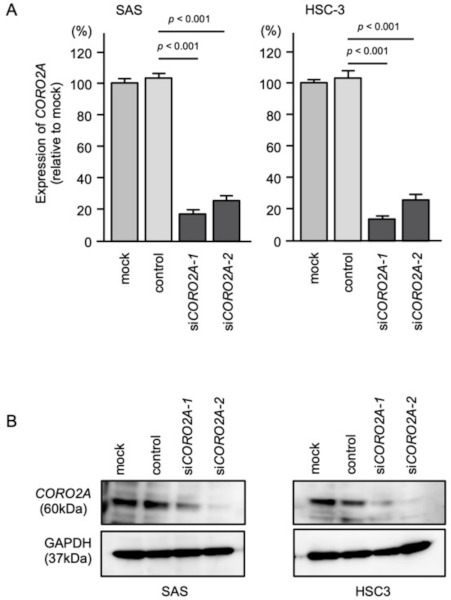
Knockdown efficiencies of siRNAs in OSCC cell lines (SAS and HSC-3 cells). Knockdown efficiencies of *CORO2A* expression by *siCORO2A-1* and *siCORO2A-2* were evaluated by real-time PCR (**A**) and Western blotting (**B**). Data of expression of CORO2A (mRNA) and *CORO2A* (protein) were collected 72 h after siRNAs transfection. GAPDH (mRNA) and GAPDH (protein) were used as internal controls.

**Figure 6 ijms-22-12684-f006:**
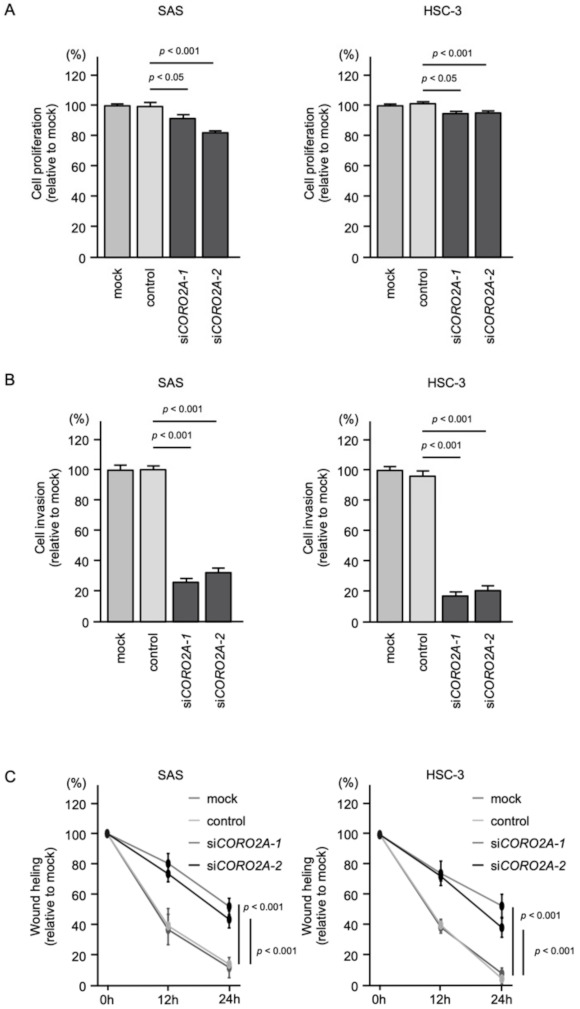
Functional assays of cell proliferation, migration, and invasion following transient transfection of siRNAs in OSCC cell lines (SAS and HSC-3 cells). (**A**) Cell proliferation was assessed using XTT assays. Data were collected 96 h after siRNAs transfection. (**B**) Cell invasion was determined 48 h after seeding miRNA-transfected cells into chambers using Matrigel invasion assays. (**C**) Cell migration was assessed with a wound-healing assay. Data were collected 0 h, 12 h, and 24 h after cell scratch.

**Figure 7 ijms-22-12684-f007:**
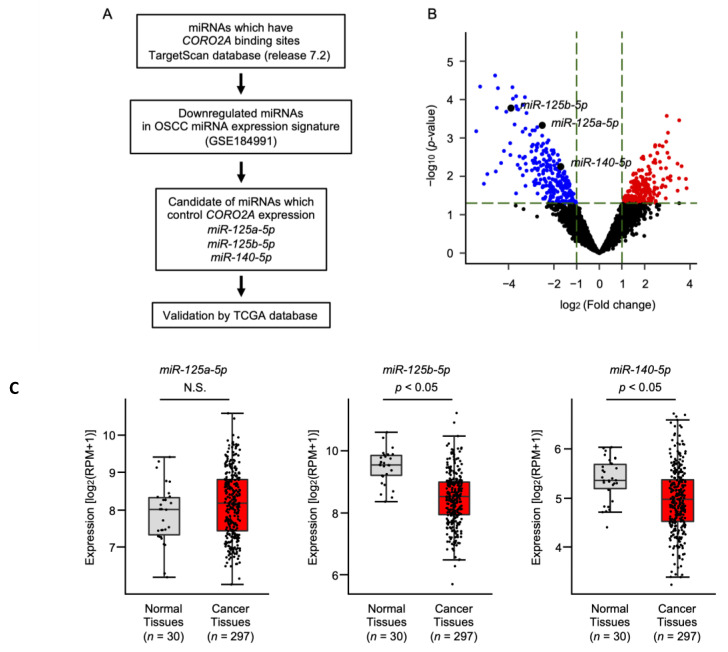
Selection of *CORO2A* controlled miRNAs in OSCC cells. To identify miRNAs controlling *CORO2A* expression in OSCC cells, we used the TargetScan database (release 7.2) and the miRNA expression signature of OSCC. (**A**) Flowchart of the strategy used to identify the candidate of *CORO2A* regulated miRNAs in OSCC cells. (**B**) Volcano plot of the miRNA expression signature determined through RNA sequencing. The log_2_ fold change (FC) is plotted on the x-axis, and the log_10_ (*p*-value) is plotted on the y-axis. The blue points represent the downregulated miRNAs with an absolute log_2_ FC <−1.0 and *p* < 0.05. (**C**) The expression levels of *miR-125a-5p*, *miR-125b-5p*, and *miR-140-5p* evaluated in an HNSCC dataset from TCGA (N.S., not significant).

**Figure 8 ijms-22-12684-f008:**
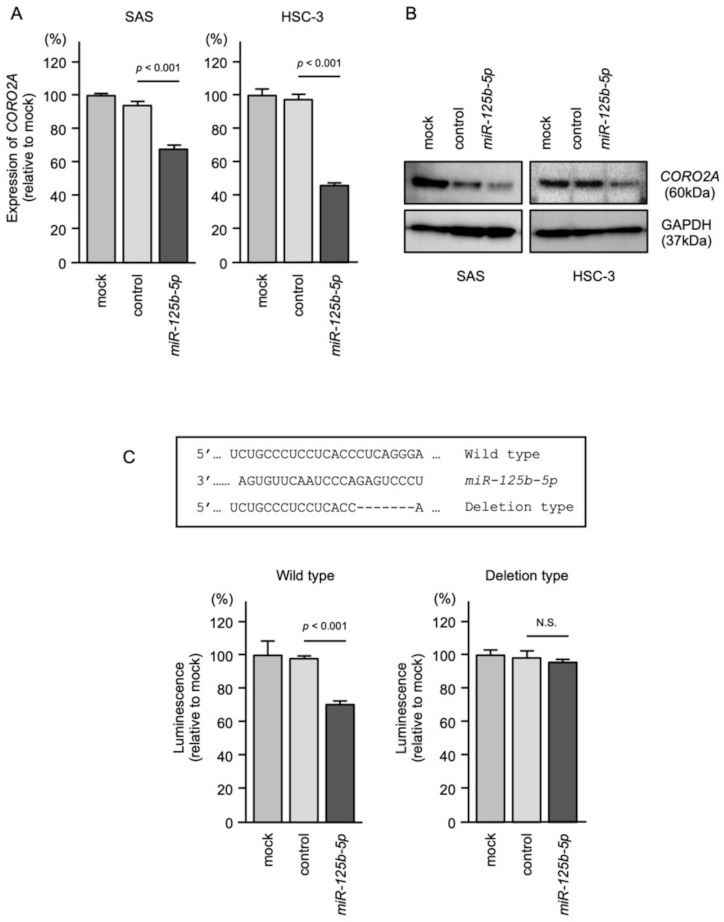
Direct regulation of *CORO2A* expression by *miR-125b-5p* in OSCC cells. (**A**) Real-time PCR showing the significantly reduced expression of *CORO2A* mRNA at 48 h after *miR-125b-5p* transfection in SAS and HSC-3 cells. Expression of GAPDH was used as an internal control. (**B**) Western blot showing reduced expression of the *CORO2A* protein at 48 h after *miR-125b-5p* transfection in SAS and HSC-3 cells. Expression of GAPDH was used as an internal control. (**C**) TargetScan database shows that a single putative *miR-125b-5p* binding site predicts the 3′UTR of *CORO2A* sequence (upper panel). Dual-luciferase reporter assays showed reduced luminescence activity after co-transfection of the wild-type vector and *miR-125b-5p* in HSC-3 cells (lower panel). Normalized data were calculated as the Renilla/firefly luciferase activity ratio (N.S., not significant).

**Figure 9 ijms-22-12684-f009:**
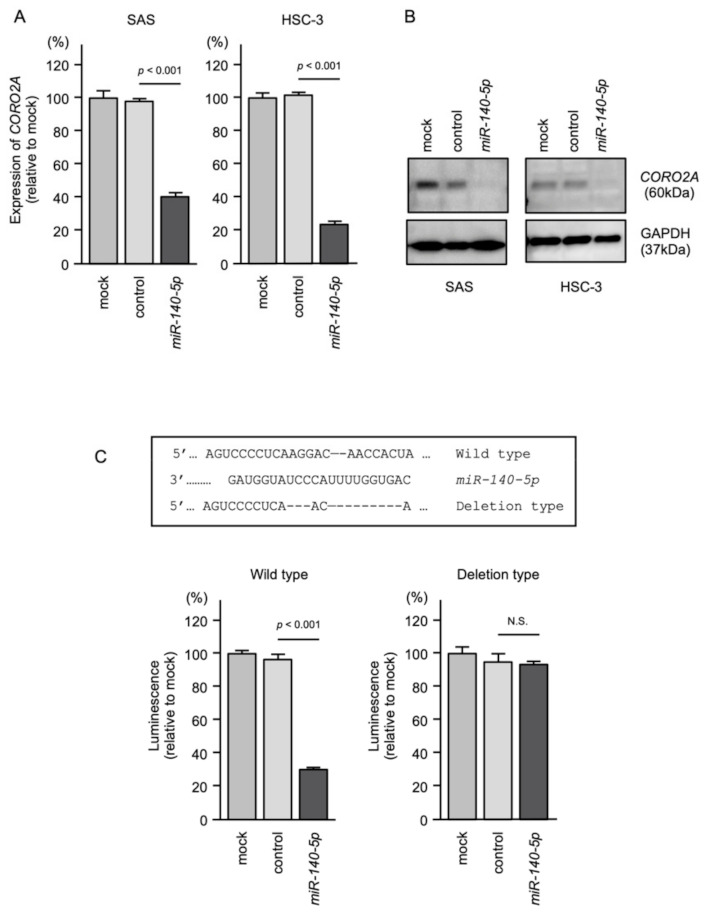
Direct regulation of *CORO2A* expression by *miR-140-5p* in OSCC cells. (**A**) Real-time PCR showing the significantly reduced expression of *CORO2A* mRNA at 48 h after *miR-140-5p* transfection in SAS and HSC-3 cells. Expression of GAPDH was used as an internal control. (**B**) Western blot showing reduced expression of the *CORO2A* protein at 48 h after *miR-140-5p* transfection in SAS and HSC-3 cells. Expression of GAPDH was used as an internal control. (**C**) TargetScan database shows that a single putative *miR-140-5p* binding site predicts the 3′UTR of *CORO2A* sequence (upper panel). Dual-luciferase reporter assays showed reduced luminescence activity after co-transfection of the wild-type vector and *miR-140-5p* in HSC-3 cells (lower panel). Normalized data were calculated as the Renilla/firefly luciferase activity ratio (N.S., not significant).

**Figure 10 ijms-22-12684-f010:**
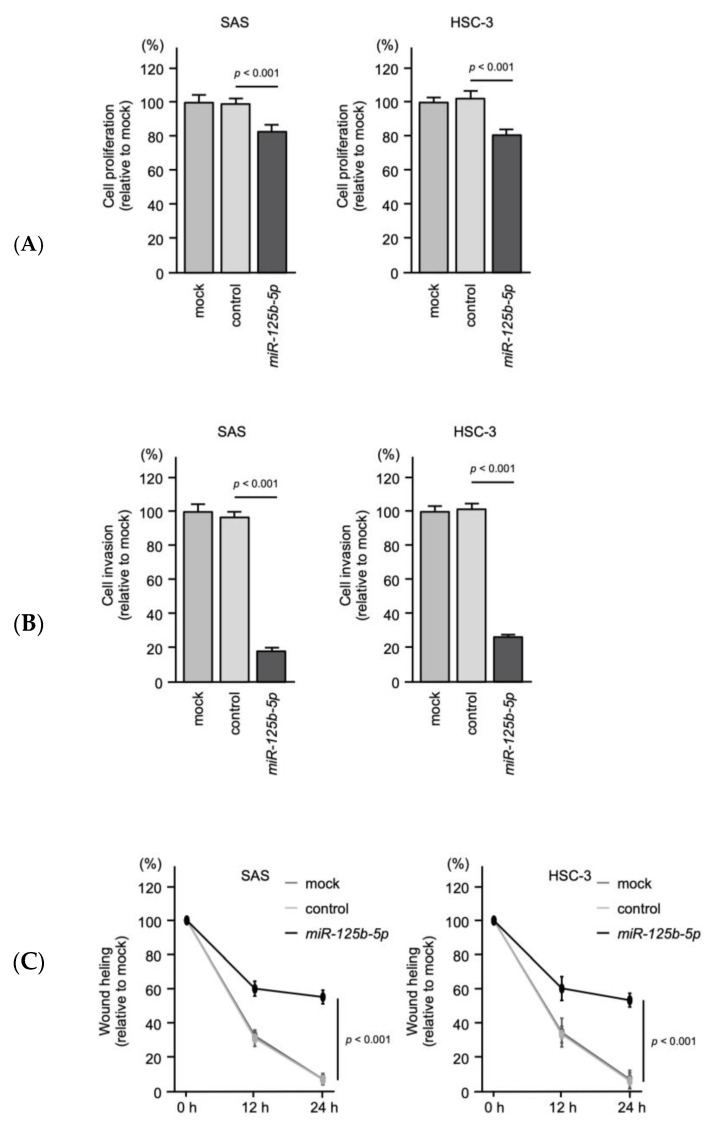
Tumor-suppressive function of *miR-125**b-5p* in OSCC cells. (**A**) Cell proliferation assessed with an XTT assay at 72 h after transfection of mature miRNAs. (**B**) Cell invasion determined with a Matrigel invasion assay at 48 h after seeding miRNA-transfected cells into the chambers. (**C**) Cell migration was assessed with a wound-healing assay. Data were collected 0 h, 12 h, and 24 h after cell scratch.

**Figure 11 ijms-22-12684-f011:**
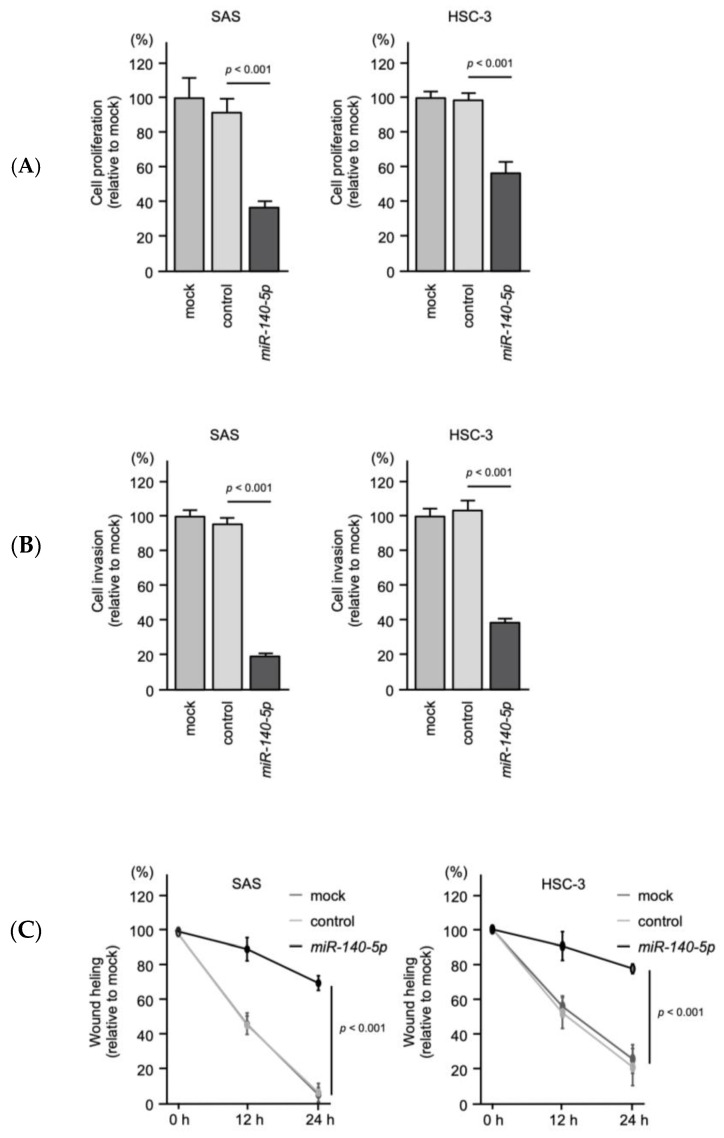
Tumor-suppressive function of *miR-140-5p* in OSCC cells. (**A**) Cell proliferation assessed with an XTT assay at 72 h after transfection of mature miRNAs. (**B**) Cell invasion determined with a Matrigel invasion assay at 48 h after seeding miRNA-transfected cells into the chambers. (**C**) Cell migration was assessed with a wound-healing assay. Data were collected 0 h, 12 h, and 24 h after cell scratch.

## Data Availability

Our expression data were deposited in the GEO database (accession number: GSE184991).

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
