# Peer review of "Molecular Pathogenesis of the Coronin Family: CORO2A Facilitates Migration and Invasion Abilities in Oral Squamous Cell Carcinoma"

_ijms, 2021, doi:10.3390/ijms222312684_

Round 1

Reviewer 1 Report

The article is interesting and reports valuable information to increase our knowledge and degree of evidence on the roles of coronin proteins in oral cancer. However, some points affect its internal validity (high risk of bias) and shuold be modified before its acceptance and publication:

Main methodological problem:

1. The comparison between cancer tissues and normal tissues could be biased due to innapropiate control group. The original source (i.e., anatomical and health status) of control group should be clarified. If authors should clarify if they considered matched epithelia close to oral carcinomas samples as “normal tissues” and  healthy control groups. Accordingly, the methodology of this work would be seriously biased (low internal validity), since the presence of genetically altered premalignant fields is well established in patients with head and neck cancer. The field cancerization theory is well known and accepted in this anatomical area, where the oral mucosa apparently clinically healthy harbors early oncogenic molecular alterations, with very important prognostic implications. For this reason, the adjacent non-tumor tissue of these patients should not be used as a control group, and It should not be considered as a "normal" or healthy sample. Therefore, the conclusions of the present work could be biased due to not having selected an adequate control group.

Other major limitations:

2. Statistical analysis section is incomplete. All analyses were not included, for example survival analysis, one of the most relevant analyses derived from the present investigation. Kaplan-Meier curves are descriptive, and log-rank test analytic (here a p-value was calculated), nevertheless, the estimation of effect sizes (in addition to p-values) is currently better accepted due to their better statistical properties and epidemiological information. Overall survival is a time-to-event variable, so authors should also report hazard ratios, with their corresponding 95% confidence intervals, preferably adjusted in a multivariable COX regression analysis (to account for potential confounding variables). Only p-values for survival analysis were calculated, it is a very limited and out of date misleading practice. Hazard ratios offers the magnitude of effect size, direction and precision, allowing direct comparisons with other prognostic biomarkers in oral cancer survival, and in summary, better translational potential for clinical practice.

3.references are out to date and some important topics were not introduced or discussed.

3.1. Molecular and clinical studies were introduced, but oral cancer epidemiology was neglected. This is inappropriate. Specific references on the epidemiology of oral cancer are highly advisable (i.e., GLOBOCAN last report: Bray F, Ferlay J, Soerjomataram I, Siegel RL, Torre LA, Jemal A. Global cancer statistics 2018: GLOBOCAN estimates of incidence and mortality worldwide for 36 cancers in 185 countries. CA Cancer J Clin. 2018;68(6):394-424. doi:10.3322/caac.21492). The authors should also improve the initial paragraph, minimally including the number of new cases and deaths per year.

3.2. There is a growing body of literature confirming the interactions between coronin members (e.g., coronin 1B) and cortactin. CTTN gene, located at the well-known oncogenic chromosomal band 11q13, encodes cortactin, an actin-binding protein which also regulates cell invasion and migration in oral cancer, singularly though interactions with Arp2/3complex. Coronin members appears to antagonize cortactin oncogenic role by directly binding to the Arp2/3 complex, inhibiting cortactin-mediated actin nucleation and destabilizing branched actin networks (source: Ramos-García P, González-Moles MÁ, González-Ruiz L, et al. An update of knowledge on cortactin as a metastatic driver and potential therapeutic target in oral squamous cell carcinoma. Oral Dis. 2019;25(4):949-971. doi:10.1111/odi.12913).

This interaction was not discussed, and could be very relevant, because CTTN/cortactin is currently considered one of the most promising prognostic biomarkers in oral cancer (according to the higher quality of evidence found in recent meta-analyses: Ramos-García P, González-Moles MÁ, Ayén Á, González-Ruiz L, Ruiz-Ávila I, Gil-Montoya JA. Prognostic and clinicopathological significance of CTTN/cortactin alterations in head and neck squamous cell carcinoma: Systematic review and meta-analysis. Head Neck. 2019;41(6):1963-1978. doi:10.1002/hed.25632).

Prognostic biomarkers have been extensively studied with a higher quality of evidence in other papers (this was a primary level study, when secondary- and tertiary-level studies have been published -i.e., systematic reviews and meta-analyses- according to evidence hierarchy offering a higher scientific level). Currently, systematic reviews and meta-analyses are the best tools to synthetize the quality of evidence in health sciences and singularly in oral oncology.

No meta-analysis was discussed in this paper (n=0). It is true that no meta-analysis has been published to date on coronin members and their roles in cancer, but other related proteins like cortactin has been investigated with a high level of evidence. So, references are out to date and important papers published on this topic were not cited. These references, potential biological roles and prognostic aspects should be integrated and discussed.

4.A better elaborated objectives paragraph should be reported. More detailed aims of the study would be advisable (e.g., PECO(S,T) format. Authors are also encouraged to raise their pre-specified hypotheses.

5.A final paragraph in the discussion section should be created performing a critical appraisal of the limitations of this study.(this is a common practice, mandatory in high impact journals, and strongly encouraged by influential reporting guidelines for all study design types: PRISMA, STROBE, CONSORT…). Recommendations for future studies should also be raised, due to the vast knowledge of authors on this topic.

Reviewer 2 Report

Reviewer's report

Title: Molecular pathogenesis of the coronin family: CORO2A facilitates migration and invasion abilities in oral squamous cell carcinoma

Version: 1 Date:  2021 November 12nd

Reviewer's report:

In this manuscript, the Authors investigated the role of the coronin family members (i.e. CORO1A, CORO1B, CORO1C, 81 CORO2A, CORO2B, CORO6, and CORO7), evaluating gene expression data from TCGA database in 315 OSCC cases and 30 normal tissues. They found that CORO1C, CORO2A, and CORO7 were significantly up-regulated, and CORO2B was down-regulated. From a clinical point of view, they also found by Kaplan-Meier survival curves that patients with high expression of CORO1B and CORO2A had worse prognosis. Aberrant expression was confirmed by IHC analysis on tissue samples. They demonstrated that knocking down CORO2A in cell lines by siRNA was correlated to the inhibition of cell invasion and migration abilities. Finally they found firstly in silico, and after by real time PCR and luciferase activity that miR-125a-5p, miR-125b-5p, and miR-140-5p regulate CORO2A expression in OSCC cells.

Overall, the findings outlined in this manuscript related to coronin family in OSCC are very interesting and this paper is very well written.

I have no revision to suggest.

Round 2

Reviewer 1 Report

Authors have taken all my comments into account. The manuscript has been improved and is now suitable for publication.

I strongly suggest the acceptation of this paper under its current form.